# Fibrinogen and Atherosclerotic Cardiovascular Diseases—Review of the Literature and Clinical Studies

**DOI:** 10.3390/ijms23010193

**Published:** 2021-12-24

**Authors:** Stanisław Surma, Maciej Banach

**Affiliations:** 1Faculty of Medical Sciences in Katowice, Medical University of Silesia in Katowice, 40-752 Katowice, Poland; stanislaw.surma@med.sum.edu.pl; 2Club of Young Hypertensiologists, Polish Society of Hypertension, 80-952 Gdansk, Poland; 3Department of Preventive Cardiology and Lipidology, Medical University of Lodz, 93-338 Lodz, Poland; 4Cardiovascular Research Centre, University of Zielona Gora, 65-417 Zielona Gora, Poland; 5Department of Cardiology and Adult Congenital Heart Diseases, Polish Mother’s Memorial Hospital Research Institute (PMMHRI), 93-338 Lodz, Poland

**Keywords:** fibrinogen, atherosclerosis, atherosclerotic cardiovascular disease

## Abstract

Atherosclerotic cardiovascular diseases (ASCVD), including coronary artery disease, cerebrovascular disease, and peripheral arterial disease, represent a significant cause of premature death worldwide. Biomarkers, the evaluation of which would allow the detection of ASCVD at the earliest stage of development, are intensively sought. Moreover, from a clinical point of view, a valuable biomarker should also enable the assessment of the patient’s prognosis. It has been known for many years that the concentration of fibrinogen in plasma increases, inter alia, in patients with ASCVD. On the one hand, an increased plasma fibrinogen concentration may be the cause of the development of atherosclerotic lesions (increased risk of atherothrombosis); on the other hand, it may be a biomarker of ASCVD, as it is an acute phase protein. In addition, a number of genetic polymorphisms and post-translational modifications of fibrinogen were demonstrated that may contribute to the risk of ASCVD. This review summarizes the current data on the importance of fibrinogen as a biomarker of ASCVD, and also presents the relationship between molecular modifications of this protein in the context of ASCVD.

## 1. Introduction

Atherosclerotic changes appear from childhood [1]. A number of risk factors, such as hyperlipidemia, hyperhomocysteinemia, arterial hypertension, hyperuricemia, smoking, metabolic syndrome, hypertriglyceridemia and diabetes, accelerate the progression of atherosclerotic lesions, leading to the development of atherosclerotic cardiovascular disease (ASCVD) [2]. ASCVD is defined as a coronary artery disease (CAD), cerebrovascular disease, or peripheral arterial disease (PAD) of atherosclerotic origin. ASCVD represents the number one cause of morbidity and mortality worldwide [3]. In 2019, the number of patients with cardiovascular diseases worldwide was 523 million, while the number of deaths due to these diseases reached 18.6 million [4]. In 2017, the number of patients with CAD worldwide reached 126 million (1.72% of the world population), and it is estimated to increase every year. Worldwide, CAD caused nine million deaths in 2017, making the disease the leading cause of death [5]. The incidence of stroke is also a significant problem. In 2019, the number of patients with stroke worldwide was 101 million, while the number of deaths due to stroke was 6.55 million [6]. PAD is also a widespread disease. In 2019, the number of patients with PAD worldwide was 113 million, and the disease caused 74.1 thousand deaths [4].

The huge prevalence of ASCVD diseases worldwide means that factors that cause their occurrence are searched for in order to develop appropriate methods of prevention and therapy. One such factor is the plasma fibrinogen concentration.

Already in the 1950s, it was found that plasma fibrinogen concentration is associated with a risk of CVD [7]. Many years ago, in the Framingham study involving 1315 healthy people, a relationship was found between plasma fibrinogen concentration and the incidence of CVD, including CAD and stroke, over a 12-year follow-up. People whose fibrinogen plasma concentration was in the second and third terciles (concentrations: 2.7–3.1 g/L and 3.1–7.0 g/L, respectively) were characterized by a significantly higher frequency of CVD than those in the first tercile (1.3–2.7 g/L). A higher plasma fibrinogen concentration, as a separate variable, had a similar effect on cardiovascular risk as well-known risk factors such as smoking cigarette, obesity, arterial hypertension, and diabetes [8].

This literature review summarizes the pathophysiological and clinical understanding of fibrinogen and its relationship with the risk of ASCVD.

## 2. Fibrinogen—Physiological and Pathophysiological Aspects

Fibrinogen, coagulation factor I, is a 340 kDa glycoprotein that plays an important role in many physiological and biochemical processes. The name “fibrinogen” was used for the first time in 1847 by Rudolf Virchow, while in 1872, Alexander Schmidt indicated that the conversion of fibrinogen to fibrin is an enzymatic process [9]. Physiologically, almost all fibrinogen is found in plasma, and its concentration is 1.5–3.5 g/L (normal concentration ranges may vary slightly among different laboratories), with a half-life (T_1/2_) of 3–5 days [10,11]. The plasma concentration of fibrinogen largely depends on the factors regulating its synthesis and genetic predisposition [10,11]. Fibrinogen is expressed primarily in hepatocytes and is regulated transcriptionally and post-transcriptionally. The biosynthesis of fibrinogen in the liver is primarily constitutive. Fibrinogen synthesis is regulated by acute-phase proteins, mainly by IL-6 (derived by monocytes, macrophages, and vascular endothelial cells), which induces its synthesis in the liver, while IL-1β and tumor necrosis factor-alpha (TNF-α) suppress its synthesis [10,12]. Thus, fibrinogen is an acute phase protein because its biosynthesis is increased during inflammation (in the acute phase of inflammation, plasma fibrinogen concentrations can exceed 7 g/L). The production of fibrinogen is also increased by glucocorticosteroids (GC) (Figure 1) [10]. 

Fibrinogen biosynthesis begins with the expression of the three genes *FGA*, *FGB* and *FGG*, clustered in a 50 kb region of human chromosome 4 (long arm; 4q31.3–4q32.1) [10]. The mechanisms regulating the expression of fibrinogen genes are not fully understood. Genome-wide association studies (GWAS) identified single-nucleotide polymorphisms (SNP) within fibrinogen genes, as well as loci distinct from fibrinogen that implicate transcription factors (e.g., hepatocyte nuclear factors 1 and 4 (HNF1 and HNF4), signal transducer and activator of transcription 3 (STAT3) and inflammatory signaling pathways downstream of IL-6 in fibrinogen expression. In addition, microRNA (miR) from the hsa-miR-29 and hsa-miR-409-3p family reduce fibrinogen expression in hepatoma cells in vitro, revealing mechanisms that can fine-tune fibrinogen levels in response to environmental signals [13]. The transcription of mRNA leads to the generation of three homologous polypeptide chains: Bβ, Aα and γ. The assembly of the six chains takes place in a stepwise manner, in which single chains assemble first into Aα/γ and Bβ/γ complexes, then into Aα/Bβ/γ half-molecules, and finally into hexameric complexes (Aα/Bβ/γ)_2_ linked by disulfide bonds (Figure 1) [11]. Fibrinogen has a complex trinodular structure with a central nodule (E-domain) that contains the N-terminus of each chain and two lateral globular domains (D-domains) that contain the C-terminus of Bβ- and γ-chains. The E-domain is linked to the two D-domains by a coiled-coil triple helix structure [14].

Post-translational modifications of fibrinogen significantly affect its biological function. A systematic review by de Vries et al. summarized the knowledge of post-translational modifications of fibrinogen. A number of post-translational modifications were identified: oxidation, nitration, glycosylation, glycation, acetylation, phosphorylation, homocysteinylation, citrullinization, carbamylation and guanidinylation. These modifications were found to affect the rate of fibrin polymerization, the blood clot structure, and the course of fibrinolysis. Thus, post-translational modifications of fibrinogen may play an important role in the physiology and pathophysiology of blood coagulation [15].

Moreover, genetic polymorphisms, which may influence the risk of various diseases, play an important role in the properties of fibrinogen [16].

Many factors and/or conditions were shown to increase the plasma concentration of fibrinogen. These include: female gender, Black ethnicity, age, diabetes, smoking and alcohol consumption, arterial hypertension, obesity, lipid disorders, metabolic syndrome, menopause, oral contraceptives, microalbuminuria, lower socioeconomic status and premature family history of CVD. The association with body mass index (BMI) was twice as strong in women as in men. However, the association with smoking cigarettes was much stronger in men. Interestingly, plasma fibrinogen concentration is inversely related to serum HDL cholesterol (high-density lipoprotein cholesterol) concentration [17,18,19]. Interestingly, the renin-angiotensin-aldosterone system (RAAS) plays an important role in the regulation of fibrinogen plasma concentration. Therefore, as pointed out by Kryczka et al., differences in the regulation of RAAS in women (including the effect of estrogens) and in men may affect the fibrinogen plasma concentration and observed clinical effect [20]. The basic physiological function of fibrinogen is participation in the final stage of the clotting process and transformation into a fibrillar protein—fibrin, which forms a blood clot (Figure 2) [11].

Moreover, fibrinogen is involved in the matrix physiology (by interaction with plasminogen, FXIII, vitronectin and fibronectin), regulation of the inflammatory process, infection, wound healing, intercellular interaction, cell migration, tumor growth, angiogenesis, and metastasis [11].

The increased fibrinogen plasma concentration directly activates many mechanisms, which, consequently, may intensify the progression of atherosclerosis [20]. The pro-atherogenic mechanisms of action of fibrinogen are summarized in Table 1.

In summary, fibrinogen has many important functions in human physiology, but also many unfavorable pathophysiological pathways induced by increased fibrinogen plasma concentration were described, which aggravate the atherosclerotic process.

## 3. Fibrinogen and Cardiovascular Risk

The results of numerous epidemiological studies indicate that increased plasma fibrinogen concentrations are a risk factor for ASCVD. Importantly, the results of some studies even indicate that increased plasma fibrinogen concentrations more adversely affect the risk of CVD than only increased serum cholesterol concentrations. In the European Concerted Action on Thrombosis and Disabilities Angina Pectoris Study (ETAT), which enrolled over 3000 patients with angiographically documented CAD, cholesterol was not found to be an independent risk factor for coronary events. Its mean concentration was 257 mg/dL in the group with adverse events in comparison to 246 mg/dL in the group of people without complications. However, a strong relationship was demonstrated in relation to fibrinogen. The risk of coronary events, including sudden cardiac death, increased with fibrinogen plasma concentrations, which remained low in the case of high cholesterol, as long as the fibrinogen concentration remained low [22]. Moreover, a prospective study by Ma et al., including 14,916 men in the Physicians’ Health Study, aged 40–84 years, assessed the relationship between the plasma fibrinogen concentration and the risk of a myocardial infraction (MI). In addition, subjects were randomized to take aspirin (325 mg every other day) or placebo for 5 years. It was shown that people who experienced an MI had a higher plasma fibrinogen concentration (*p* = 0.02). A high fibrinogen plasma concentration (≥343 mg/dL) had a twofold increase in MI risk (RR = 2.09; 95% CI: 1.15–3.78) compared with those with fibrinogen below 343 mg/dL. There was no interaction between the fibrinogen plasma concentration and aspirin treatment. Thus, it was found that fibrinogen was associated with an increased risk of future MI independent of other CVD risk factors, atherogenic factors such as lipids, and antithrombotics such as aspirin [23].

Table 2 summarizes the studies that assessed the relationship between plasma fibrinogen concentrations and CVD risk in different patients’ groups.

The results of the studies summarized in Table 1 confirm that the increased fibrinogen plasma concentration is a risk factor for CVD and worsens the prognosis of patients. Meta-analyses from previous studies also confirmed these observations. A meta-analysis by Danesh et al., including the results of 31 prospective studies, showed that each 1 g/L increase in plasma fibrinogen was associated with an increase by 142% (HR = 2.42; 95% CI: 2.24–2.60) for the risk of CAD (fatal or non-fatal), by 106% (HR = 2.06; 95% CI: 1.83–2.33) for the risk of stroke (fatal or non-fatal), by 176% (HR = 2.76; 95% CI: 2.28–3.35) for the risk of other vascular deaths, and by 103% (HR = 2.03; 95% CI: 1.90–2.18) for the risk of nonvascular deaths. The observed effects were independent of age and gender and remained significant after adjusting for other cardiovascular risk factors [45]. An analysis of the results of 52 prospective studies by Kaptoge et al. assessed the predictive value of plasma fibrinogen concentration in relation to the risk of CVD. The analysis included 246,669 people without previously diagnosed CVD. It was shown that an increased fibrinogen plasma concentration was associated with a 15% higher risk of CVD (HR = 1.15; 95% CI: 1.13–1.17). Researchers indicated that the assessment of plasma fibrinogen concentration may be particularly advantageous in patients with an intermediate CVD risk [46]. A meta-analysis of 23 studies by Song et al. included 2984 CAD cases and 2279 controls, assessing the relationship between plasma fibrinogen concentration and the risk of CAD. It was shown that patients with CAD had significantly higher plasma fibrinogen concentrations compared to patients without CAD (*p* < 0.0001). Moreover, the predicted odds ratio for a 1 g/L higher plasma fibrinogen concentration was 0.94 (95% CI: 0.78–1.10). Furthermore, fibrinogen plasma concentrations were slightly related to age-related CAD patients. Therefore, the plasma fibrinogen concentrations were correlated with CAD and may be a risk factor and predictor of CAD [47].

Additionally, the data suggesting that plasma fibrinogen concentration are of clinical interest and may be a useful biomarker for subclinical atherosclerosis. 

In a study of 652 men aged 40–60 years old (asymptomatic and never treated from CVD causes) with at least one of the following CVD risk factors: total cholesterol >6.2 mmol/L and/or systolic blood pressure ≥160 mmHg and/or diastolic blood pressure ≥95 mmHg, and/or smoking cigarettes, the relationship between plasma fibrinogen concentration and the presence of atherosclerotic plaques was assessed. The independent associations between fibrinogen and the presence and extent of atherosclerosis were indicated. Plaque prevalence was significantly more pronounced with the increasing tercile of fibrinogen plasma concentrations. The odds ratio of the upper to lower fibrinogen terciles for the presence of plaque was 1.6 (95% CI: 1.4–1.8) and 1.4 (95% CI: 1.2–1.7) for its extent. The adjustment for other risk factors slightly reduced the association between fibrinogen and atherosclerosis without changing the direction of the associations [48]. In the newer CARDIA study (Coronary Artery Risk Development in Young Adults) by Green et al. also assessed the possibility of using fibrinogen as a biomarker of subclinical atherosclerosis. The study included 1396 participants aged 25–37 who were assessed for coronary artery calcification (CAC) and carotid intimal/medial thickness (CIMT) 13 years after enrolment in the study. It was shown that the prevalence values of CAC with increasing quartiles of fibrinogen were 14.4%, 15.2%, 20.0%, and 29.1% (*p* < 0.001). This finding was still significant after adjusting for a number of risk factors for CVD. A similar trend was observed for CIMT (*p* = 0.014). Interestingly, the prevalence of CAC was not associated with increasing quartiles of FVII, FVIII, or the von Willebrand factor, suggesting they may be less involved in plaque progression than fibrinogen. The researchers concluded that the elevated fibrinogen plasma concentration in persons aged 25–37 was independently associated with subclinical cardiovascular disease in the subsequent decade [49]. Other analyses of the 13-year CARDIA study also found that higher fibrinogen plasma concentrations during young adulthood were positively associated with the incidence of CAC and increased CIMT in middle age [50]. Moreover, in a study by Menti et al., involving 74 people with excess weight and mild, untreated dyslipidemia, the possibility of using fibrinogen as a biomarker of a vascular endothelial function using brachial artery flow-mediated dilation (BAFMD) was assessed. Higher serum concentrations of fibrinogen were shown to be significantly and independently associated with a BAFMD below 8% (*p* = 0.02) [51].

The research results also provided interesting data indicating the possibility of using fibrinogen as a biomarker for atherosclerotic plaque composition. Patients with a non-calcified plaque (NCP) or mix plaque (MP) are known to have a higher risk of poor CVD outcomes. In a study by Li et al. involving 329 people, the relationship between plasma fibrinogen concentration and NCP and MP was assessed. It was shown that female patients with NCP/MP had significantly higher fibrinogen plasma concentrations compared to male patients. A multiple logistic regression analysis showed that higher fibrinogen plasma concentrations were an independent risk factor for the presence of NCP/MP (OR = 3.677, 95% CI: 1.539–8.785, *p* = 0.003) in females (optimal plasma fibrinogen concentrations cut-off value for the NCP/MP prediction was 3.41 g/L) [52]. However, a study by Wang et al., involving 154 patients with CAD, and using intravascular optical coherence tomography, showed that serum fibrinogen concentration was not associated with coronary atherosclerotic plaque vulnerability [53].

More and more data appear in the literature indicating the great usefulness of the plasma fibrinogen concentration test in combination with other biomarkers (D-dimer and albumin) for the assessment of CVD risk. 

A very interesting study by Bai et al. assessed the clinical usefulness of the D-dimer to fibrinogen ratio (DFR) in patients with CAD after percutaneous coronary intervention (PCI). The patients were divided into two groups according to DFR values: the lower group (DFR < 0.52, n = 2123) and the higher group (DFR ≥ 0.52, n = 1073). The follow-up time was 37.59 ± 22.24 months, and the primary endpoints were all-cause mortality (ACM) and cardiac mortality (CM). Significant differences between the two groups in terms of ACM (2.4% vs. 6.6%, *p* < 0.001) and CM (1.5% vs. 4.0%, *p* < 0.001) were shown. DFR was found to be an independent predictor of ACM (HR = 1.743; 95% CI: 1.187–2.559, *p* = 0.005) and CM (HR = 1.695; 95% CI: 1.033–2.781, *p* = 0.037) in long-term follow-up for patients with CAD after PCI [54]. 

A number of studies indicate a high diagnostic potential for the determination of the fibrinogen to albumin ratio (FAR). In the study by Zhang et al., including 5829 patients with CAD after PCI, the clinical value of FAR in predicting CVD events was assessed. Patients were divided according to FAR values (FAR < 0.095, n = 3811), and a high group (FAR ≥ 0.095, n = 2018) were followed for 35.9 ± 22.6 months. FAR was shown to be independently correlated with all-cause mortality (HR = 1.432; 95% CI: 1.134–1.808, *p* = 0.003), cardiac mortality (HR = 1.579; 95% CI: 1.218–2.047, *p* = 0.001), major adverse cardiac and cerebrovascular events (HR = 1.296; 95% CI: 1.125–1.494, *p* < 0.001), major adverse cardiac events (HR = 1.357; 95% CI: 1.170–1.572, *p* < 0.001), and heart failure (HR = 1.540; 95% CI: 1.135–2.091, *p* = 0.006) in long-term follow-up for patients with CAD after PCI [55]. Importantly, the high clinical usefulness of the FAR assessment was also demonstrated in the study by Roth et al., which included 344 patients with cardiogenic shock refractory who underwent veno-arterial extracorporeal membrane oxygenation (VA-ECMO). It was shown that a higher FAR was significantly associated with the risk of in-hospital thromboembolic complications (OR = 3.72; 95% CI: 2.26–6.14) [56]. The FAR assessment may also be useful in assessing the severity of CAD. A study by Karahan et al., involving 278 patients with STEMI, assessed the relationship between FAR and extent and severity of CAD evaluated by TAXUS Drug-Eluting Stent Versus Coronary Artery Bypass Surgery for the Treatment of Narrowed Arteries (SYNTAX) Score (SS). A significant association was demonstrated between FAR and SS (r = 0.458, *p* < 0.001), stating that FAR was significantly related to SS in predicting the severity of CAD in patients with STEMI [57]. The study by Celebi et al. also showed that FAR was significantly associated with the severity of CAD in patients with stable CAD assessed based on the SS scale [58]. The study by Zhao et al. assessed the clinical usefulness of FAR at admission for predicting the spontaneous recanalization of the infarct-related artery (IRA) in 255 patients with STEMI. FAR was shown to be significantly lower in the spontaneous recanalization group than in the non-spontaneous recanalization group (*p* < 0.001). FAR was negatively correlated with the spontaneous recanalization of the infarct-related artery in patients with acute STEMI (OR = 0.492; 95% CI: 0.354–0.686, *p* < 0.001) [59]. A study by Erdoğan et al. assessed the clinical usefulness of FAR in predicting the SYNTAX score in 330 patients with NSTEMI. FAR was shown to be an independent predictor of the intermediate-high SYNTAX scores (OR = 1.478; 95% CI: 1.089–2.133, *p* = 0.002) [60]. The study by Li et al. also analysed the usefulness of FAR in the assessment of the long-term prognosis in 1138 NSTEMI patients first implanted with drug-eluting stent (DES). The severity of CAD was evaluated using the Gensini score (GS). The endpoints were major adverse cardiovascular events (MACE), including all-cause mortality, myocardial reinfarction, and target vessel revascularization (TVR). It was shown that FAR was an independent predictor of severe CAD (OR = 1.060; 95% CI: 1.005–1.118, *p* < 0.05) and that FAR was an independent prognostic factor for MACE at 30 days, 6 months, and 1 year after DES implantation (HR = 1.095; 95% CI: 1.011–1.186, *p* = 0.025; HR = 1.076; 95% CI: 1.009–1.147, *p* = 0.026; HR = 1.080; 95% CI: 1.022–1.141, *p* = 0.006) [61]. From a clinical point of view, the results of the study by Wang et al. are also important, showing that FAR was independently associated with the occurrence of post-contrast acute kidney injury (PC-AKI) and could significantly improve PC-AKI prediction over the Mehran risk score in patients undergoing elective PCI [62].

The potential diagnostic role of the determination of plasma fibrinogen concentration along with hsCRP or haemoglobin is also indicated [63,64].

It should be emphasized that in the coagulation, platelet is activated, and fibrinogen in plasma is clotting. Therefore, the fibrinogen concentrations in plasma and serum might have different clinical implication for diagnosis, however data on this is still inconsistent. Therefore, when using the determination of fibrinogen as an indicator of CVD risk, one should pay attention to the method of measurement. Plasma level is the preferred method for the determination of fibrinogen concentration. In summary, an increased plasma fibrinogen concentration has, for many years, been considered a biomarker of cardiovascular risk—both in people with and without CVD, as indicated by a number of observational studies and their meta-analyses. Moreover, the assessment of plasma fibrinogen concentration may be a biomarker of subclinical atherosclerosis. Currently, more and more studies indicate an even greater diagnostic importance of the assessment of DFR and FAR. However, it should be remembered that the results of observational studies can only help to formulate a research hypothesis but cannot confirm a cause-and-effect relationship. Some discrepancies between studies may result from several factors affecting plasma fibrinogen concentration as well as directly on the risk of CVD, such as smoking [65], insertion/deletion polymorphism of the angiotensin converting enzyme gene [66] and polymorphisms of the fibrinogen gene [16].

## 4. Fibrinogen Molecular Modifications and Cardiovascular Risk

A number of molecular modifications, such as gene polymorphisms, alternative splicing and post-translational modifications, may influence the plasma concentration of fibrinogen and its biochemical properties, and consequently the risk of CVD. As mentioned earlier, fibrinogen biosynthesis begins with the expression of the three genes *FGA*, *FGB* and *FGG*. Many studies indicate the significant role of modifications at the stage of fibrinogen biosynthesis in shaping the risk of CVD.

An alternatively-spliced form of the fibrinogen is γ′ fibrinogen (Figure 3) [67,68,69]. 

Fibrinogen is a disulfide-bonded dimer, with each half of the dimer containing one Aα chain, one Bβ chain, and one γ chain, which can be either the more common γA chain or a γ′ chain. γ′ fibrinogen consists of approximately 90% heterodimers containing one γ′ chain and one γA chain, and about 10% homodimers containing two γ′ chains; γ′ fibrinogen constitutes about 3–40% of total fibrinogen plasma concentrations [70,71]. In particular, compared to the total fibrinogen, γ′ fibrinogen forms fibrin blood clots that show differences in clot architecture, are mechanically stiffer, and are very resistant to fibrinolysis [67]. The review of the literature by de Willige et al. indicates a number of different roles of the fibrinogen γ′ chain in haemostasis [68]. 

The results of previous studies suggested that higher plasma concentrations of γ′ fibrinogen led to the formation of blood clots that are very resistant to fibrinolysis, which increased the risk of CVD [72,73]. In a study by Mannila et al., including 387 postinfarction patients and 387 healthy individuals, it was shown that elevated plasma γ′ fibrinogen concentration was an independent predictor of myocardial infraction (OR = 1.24; 95% CI: 1.01–1.52) [74]. Other studies, such as the study by Lovely et al. of 3042 participants from the Framingham Heart Study Offspring Cohort, found that γ′ fibrinogen plasma concentration was associated with prevalent CVD (*p* = 0.02), although the top 2 SNPs (rs7681423 and rs1049636) associated with γ′ fibrinogen plasma concentration were not associated with a risk of CVD [67]. There are also newer studies, the results of which indicate a significant role of the FGG polymorphism in the risk of CVD. In a study by Drizlionok et al. A/A and G/A genotype carriers of an SNP in rs2066865 in the FGG had a higher plasma fibrinogen concentration, and this might be associated with an increased risk of microvascular thrombosis [75].

In a study by Appiah et al. covering adults ≥65 years (n = 3219) enrolled in the Cardiovascular Health Study, the association of plasma γ′ fibrinogen concentration with the incidence of CVD, independent of established CVD risk factors and total fibrinogen, was assessed. Hazard ratio per 1 standard deviation (10.7 mg/dL) increment of γ′ fibrinogen was 1.02 (95% CI: 0.95–1.10) for CAD; 0.88 (0.77–1.00) for ischemic stroke; 1.07 (0.87–1.32) for PAD; 1.00 (0.92–1.08) for heart failure and 1.01 (0.92–1.10) for CVD mortality [76]. Nonetheless, other results were obtained in the Mendelian randomization study by Maners et al., which assessed γ′ and the total fibrinogen plasma concentration in relation to the risk of venous thromboembolism and ischemic stroke. It was shown that estimates based on a combination of 16 genetic instruments for γ′ fibrinogen and 75 genetic instruments for total fibrinogen indicated a protective effect of higher γ′ fibrinogen and higher total fibrinogen on venous thromboembolism risk. There was also a protective effect of higher γ′ fibrinogen plasma concentration on cardioembolic and large artery stroke risk [77].

Therefore, whether or not γ′ fibrinogen is simply a marker of CVD or a prospectively defined risk factor for CVD remains controversial. It is known that other factors also influence the structure of a blood clot. Clot structure may contribute to an increased CVD risk in vivo through associations with other CVD risk factors (age, metabolic syndrome, CRP, high density lipoprotein HDL cholesterol and homocysteine) independent from total or γ ‘fibrinogen plasma concentrationa [78]. Similar conclusions were reached by Pieters et al., who stated that CVD risk factors (excluding fibrinogen) explained 20% and 3%, respectively, of the variance in fibrinogen γ′ and the γ/total fibrinogen ratio, with C-reactive protein making the biggest contribution. More than 50% of the variance in fibrinogen γ′ and γ′/total fibrinogen ratio is explained by factors other than total fibrinogen or other traditional CVD risk factors [79]. Recently, Rautenbach et al. found that the iron metabolism activity may affect the plasma fibrinogen concentration and the percentage of γ′ fibrinogen [80]. Moreover, another study by Rautenbach et al. also found that alcohol intake influenced the percentage of γ′ fibrinogen, as well as modulated the influence of fibrinogen SNPs on total fibrinogen plasma concentrations [81]. The Atherosclerosis Risk in Communities (ARIC) study by Appiah et al. found that γ′ fibrinogen plasma concentrations seemed to reflect the general inflammation that accompanies and may contribute to ASCVD, instead of γ′ fibrinogen being a causal risk factor [70]. The protective effect of γ′ fibrinogen described in some studies may be due to the different roles of fibrinogen, aside from the formation of fibrin clots, in thrombotic diseases of various aetiologies [77]. The results of in vivo studies indicate that the elevated levels of the γA/γA fibrinogen isoform promote arterial thrombosis in vivo, whereas the γA/γ′ isoform does not [82]. Moreover, a significant influence on the relationship between γ′ fibrinogen and the risk of CVD may depend on the presence of various polymorphisms, such as *FGG* 9340T and *FGA* 2224G, for example [74]. In conclusion, the question of the influence of the isoform of γ′ fibrinogen on the risk of CVD requires further research.

The results of studies assessing the impact of *FGB* polymorphisms are inconsistent [83]. In a meta-analysis by Li et al., including the results of 18 case–control studies (3033 venous thromboembolism cases and 4547 healthy subjects), the relationship between the β-fibrinogen polymorphisms, -455 G/A (rs1800790) and -148 C/T, and venous thromboembolism risk was assessed. *FGB* -455 G/A and -148 C/T polymorphisms were not significantly associated with the susceptibility to venous thromboembolism in overall populations. Moreover, the -455 G/A polymorphism was associated with a decreased risk of venous thromboembolism, found among the Caucasian population [84]. In turn, the meta-analysis by Luo et al., including the results of 49 clinical trials, assessed the relationship between the same *FGB* polymorphisms with the risk of ischemic stroke. The -148 C/T polymorphism was significantly correlated with the risk of ischemic stroke in both Asians and Caucasians, while the -455 G/A polymorphism was only significantly correlated with the risk of ischemic stroke in Asians. Moreover, it was found that both *FGB* -148 C/T and -455 G/A polymorphisms were significantly correlated with the risk of cerebral infarction [85]. In a study by Canseco-Avila et al., covering 118 subjects with unstable and stable CAD, it was shown that fibrinogen plasma concentrations (>465 mg/dL) were significant in patients with CAD. These fibrinogen plasma concentrations were associated with CVD mortality during the follow-up analysis of the unstable coronary disease group (*p* = 0.04). It was also shown that the allelic loads of -455 G/A and -148 C/T were associated with plasma fibrinogen concentrations >450 mg/dL (*p* < 0.003 and *p* = 0.03) and with a risk of CAD (*p* = 0.016 and *p* < 0.006). Moreover, the follow-up of posterior events after an acute coronary event showed that the genetic load of the -148 C/T allele was associated with major adverse cardiovascular events (RR = 1.8; 95% CI: 1.01–3.35, *p* = 0.04) [86]. A meta-analysis of 26 clinical trials, by Gu et al., also assessed the association between the -455 G/A polymorphism and the risk of ischemic stroke. It was shown that this polymorphism was associated with the risk of ischemic stroke in overall, Asian, and adult analyses, but statistical association was not observed for Caucasians and children [87]. An interesting meta-analysis of the results of 45 clinical trials, by Gu et al., attempted to summarize the data on the influence of the -455 G/A polymorphism on the risk of ischemic stroke and CAD. It was shown that the -455 G/A polymorphism was associated with the risk of ischemic stroke when compared with the dominant model (OR = 1.518; 95% CI: 1.279–1.802 for AA + GA vs. GG). In the subgroup analysis by ethnicity, significantly elevated risks were associated with the A allele in Asians (OR = 1.700; 95% CI: 1.417–2.040) but not in Caucasians (OR = 0.942; 95% CI: 0.813–1.091). Moreover, it was found that the -455G/A polymorphism was associated with CAD (OR = 1.802; 95% CI: 1.445–2.246) [88]. In the study by Hu et al., the impact of the -455 G/A polymorphism on the risk of cardioembolic stroke (CES) in patients with atrial fibrillation (AF) and a low CHA_2_DS_2_-VaSc score was assessed. The study included 479 AF patients with CES and 580 AF patients without CES. It was shown that patients with the -455 G/A polymorphism had a higher plasma fibrinogen concentration (3.29 ± 0.38 mg/dL vs. 2.87 ± 0.18 mg/dL, *p* < 0.001). Moreover, it was found that -455 G/A was independently associated with an increased risk of CES in AF patients and the significance remained after the Bonferroni correction in the additive (AA vs. GA vs. GG), dominant (AA + GA vs. GG), and recessive models (AA vs. GA + GG), with ORs of 1.548 (95% CI: 1.251–1.915, *p* = 0.001), 1.588 (95% CI: (1.226–2.057, *p* = 0.003), and 2.394 (95% CI: 1.357–4.223, *p* = 0.015) [89]. In the study by Golenia et al., which included 426 Polish patients with ischemic stroke, it was shown that the β-fibrinogen -455 G/A gene polymorphism was not a risk factor for this disease [90]. Reviewing the results of other studies, one can find evidence of a different effect of the presence of the -455 G/A polymorphism on the risk of CAD (increased risk, no effect and protective effect) [91,92,93]. In a study by Martiskainen et al., covering 486 stroke patients (55–85 years), who were subjected to clinical and MRI examinations and followed over 12.5 years, the impact of the -455 G/A polymorphism on the prognosis after stroke was assessed. It was shown that women aged 55–71 years who carried the FGB -455 A-allele had worse survival regardless of smoking status compared to non-smoking FGB -455 GG homozygotes (non-smokers: HR = 5.21; 95% CI: 1.38–19.7; smokers: HR = 7.03; 95% CI: 1.81–27.3). Such a relationship was not demonstrated for women in the oldest age-group, nor among men [94].

The issue of the influence of the *FGB* polymorphism on the risk of lower extremity deep venous thrombosis (LEDVT) is also interesting. In a study by Han et al. involving 120 LEDVT patients and 120 healthy people, the relationship between six SNPs in the *FGB* promoter was assessed: -148 C/T, -249 C/T, -455 G/A, -854 G/A, -993 C/T and -1420 G/A and the risk of LEDVT. A higher fibrinogen plasma concentration was shown to increase the risk of LEDVT. The risk of LEDVT increased by 4.579 times for every unit increase in fibrinogen plasma concentration. It was found that polymorphisms such as -1420 (AG + AA) and -148 (CT + TT) were associated with a higher risk of LEDVT [95].

It is worth mentioning that congenital fibrinogen deficiencies are rare bleeding disorders characterized by extensive genetic heterogeneity in all the three genes: *FGA*, *FGB*, and *FGG* (encoding the Aα, Bβ and γ chain, respectively). Depending on the type and site of mutations, congenital defects of fibrinogen can result in variable clinical manifestations, which range from asymptomatic conditions to the life-threatening bleeding or even thromboembolic events [96,97,98]. 

The influence of the more important mutations and polymorphisms of fibrinogen genes on the risk of CVD is summarized in Table 3.

In conclusion, the influence of molecular modifications of fibrinogen, especially the polymorphisms of its genes, on the risk of CVD remains inconsistent. One possible explanation for the discrepancy in test results may be the complexity of the regulation of fibrinogen phenotype formation. A study by Cronjé et al. found that, apart from SNPs in the fibrinogen (*FGA*, *FGB* and *FGG*) genes, the fibrinogen phenotypes were also associated with SNPs in genes playing a role in lipid homeostasis (*LDL-R*, *PCSK-9*), together with *CBS* and *CRP* polymorphisms (particularly, CRP-rs3093068) [99]. Cronjé et al. also indicated the significant effect of IL-6 on the plasma fibrinogen concentration and the properties of the clot. Moreover, they highlight the possible interactions with modulating factors and that SNP effects seem to be additive and should be taken into account [100]. Similar conclusions were reached by Titov et al., who found associations between ischemic stroke and allele/genotype combinations of genes *IL6, FGA* and *FGB*, in which *IL6* plays key role and *FGA* and *FGB* have a modulating function [101]. A Mendelian randomization study by Ward-Caviness et al. found a small causal effect of fibrinogen on CAD [102]. The influence of molecular modifications of fibrinogen and the polymorphisms of its genes requires further research. Importantly, from a clinical point of view, polymorphisms of fibrinogen genes may modulate the effectiveness of CVD treatment. A study by Lynch et al., involving patients with arterial hypertension, showed that, compared to the allele homozygous for common alleles, the carrier of the -455 G/A polymorphism was found to perform better when randomized to lisinopril than chlorthalidone (for mortality and end-stage renal disease) or amlodipine (for death and stroke) [103]. 

## 5. Conclusions

ASCVD is defined as CAD, cerebrovascular disease, or PAD of atherosclerotic origin. ASCVD represents the number one cause of morbidity and mortality worldwide. The results of numerous studies indicate that plasma fibrinogen concentration, especially in conjunction with the assessment of the concentration of D’dimer or albumin, is a valuable biomarker of the primary and secondary risk of CVD. The influence of molecular modifications and genetic polymorphisms of fibrinogen genes on the risk of CVD remains controversial. Some studies showed that increased plasma fibrinogen concentrations increase the risk of CVD. Perhaps the reduction in plasma fibrinogen concentration by angiotensin-converting enzyme inhibitors, statins, fibrates or exercise is one of the factors explaining their cardioprotective effect [103,104,105]. 

Due to the ongoing COVID-19 pandemic, it should be mentioned that fibrinogen plays an important role in the pathogenesis of damage to the vascular endothelium, hypercoagulation and thrombosis in the course of this disease [106]. Perhaps the reduction in fibrinogen concentration by statins is one of the elements of their beneficial influence on improving the prognosis of COVID-19 [107,108]. Moreover, the results of numerous clinical studies indicate that the assessment of fibrinogen concentration may be a useful indicator in the assessment of the prognosis of patients with COVID-19 [109,110,111,112]. 

## Figures and Tables

**Figure 1 ijms-23-00193-f001:**
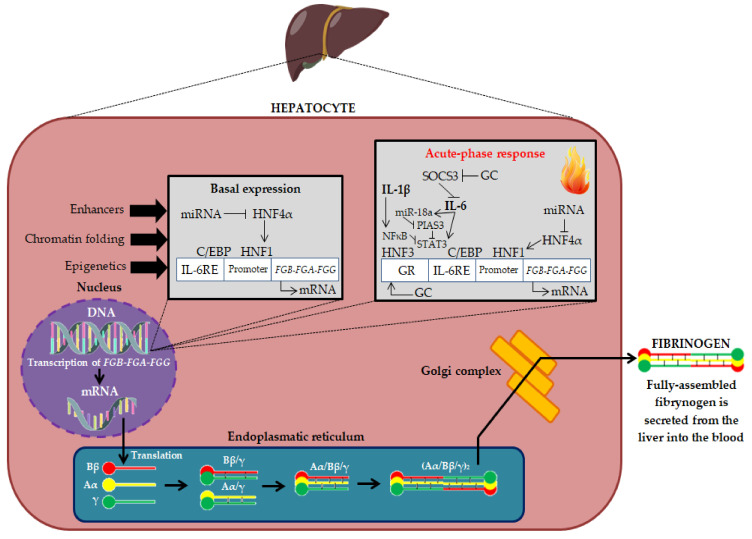
Fibrinogen biosynthesis in liver hepatocytes [9,10,11]. DNA—deoxyribonucleic acid; mRNA—Messenger ribonucleic acid; FGA-FGB-FGG—Fibrinogen chain genes; miRNA—Micro RNA; HNF4α—Hepatocyte Nuclear Factor 4 Alpha; HNF1—Hepatocyte nuclear factor 1; C/EBP—CCAAT enhancer binding proteins; IL-6RE—IL-6 response element; SOCS3—Suppressor of cytokine signaling 3; miR-18a—microRNA-18a; NFκB—Nuclear factor kappa-light-chain-enhancer of activated B cells; PIAS3—Protein inhibitor of activated STAT 3; STAT3—Signal transducer and activator of transcription 3; HNF3—Hepatocyte nuclear factor 3; IL-1β—Interleukin 1β, IL-6—Interleukin 6; GR—Glucocorticoid receptor; GC—Glucocorticosteroids; Bβ, Aα and γ—Fibrinogen polypeptide chains.

**Figure 2 ijms-23-00193-f002:**
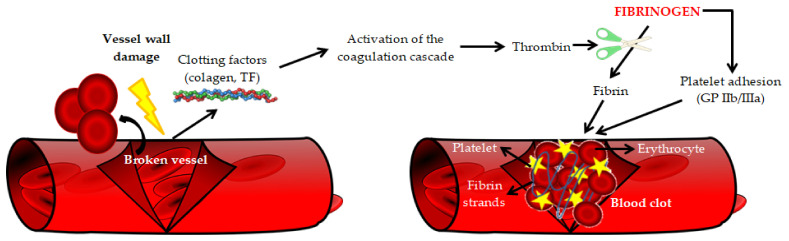
The role of fibrinogen in blood coagulation [21]. TF—Tissue factor, GP IIb/IIIa—Integrin receptors.

**Figure 3 ijms-23-00193-f003:**
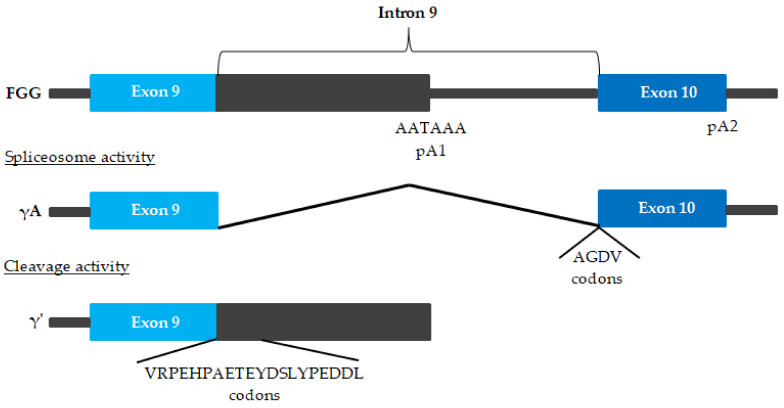
Alternative FGG pre-mRNA processing [67,68,69]. The γA chain arises when polyadenylation occurs at polyadenylation signal 2 (pA2) downstream of exon 10, and all 9 introns are removed. The alternative chain arises after polyadenylation at pA1 in intron 9. This leads to the translation of a polypeptide with a unique 20-amino acid extension encoded by intron 9, substituting the 4 γA amino acids of exon 10.

**Table 1 ijms-23-00193-t001:** The pro-atherogenic mechanisms of action of fibrinogen [20]. IL-1—Interleukin 1; TNF-α—Tumor necrosis factor α; IL-8—Interleukin 8; MCP-1—Monocyte chemoattractant protein-1; GP IIb/IIIa—Integrin receptors; IL-1β—Interleukin aβ; ICAM-1—Intercellular adhesion molecule 1; EC—Endothelial cells; LDL—Low-density lipoprotein; SMC—Smooth muscle cell.

Main Pro-Atherogenic Properties of Fibrinogen
✓ ↑ Severity of inflammation: promotes an inflammatory response by inducing the exposition of proinflammatory cytokines on monocytes (IL-1 and TNF-α) as well as chemokines, such as IL-8 and MCP-1, on endothelium and fibroblasts, which promote atherosclerotic plaque formation; ✓ Activation of platelets (via GP IIb/IIIa receptors) leading to the production of the pro-inflammatory cytokines, CD40 ligand and IL-1β, which promote atherosclerotic plaque formation; ✓ ↑ Expression of adhesion molecules (ICAM-1) on vascular EC leading to the adhesion of leukocytes, macrophages and platelets; ✓ ↑ Production of vasoactive factors by EC leading to an increase in its permeability and impairing its vasorelaxant properties; ✓ Accumulation of fibrinogen in the vessel wall enhances the infiltration of macrophages, which are precursors of foam cells; ✓ The accumulated fibrinogen deposits in the vessel wall absorb LDL cholesterol, which leads to the formation and growth of atherosclerotic plaque; ✓ Increasing the adhesion of neutrophils to activated platelets attached to the injured arterial wall, which promotes the formation of atherosclerotic plaque; ✓ ↑ SMC migration and proliferation, as well as stimulation of angiogenesis.

**Table 2 ijms-23-00193-t002:** Plasma fibrinogen concentration and CVD risk and prognosis in various groups of patients. CAD—Coronary artery disease; PCI—Percutaneous coronary intervention; 95% CI—95% confidence interval; HbA1C—Glycated hemoglobin; FBG—Fasting blood glucose; DM—Diabetes mellitus; HR—Hazard ratio; HF—Heart failure; PAD—Peripheral arterial disease; OR—Odds ratio; ICU—Intensive care unit; NSTEMI—Non-ST-segment elevation myocardial infarction; MACEs—Major adverse cardiovascular events; ACS—Acute coronary syndrome; MI—Myocardial infraction; SS—SYNTAX score; T2DM—Type 2 diabetes mellitus; CVD—Cadiovascular disease; SCD—Sudden cardiac death; GS—Gensini score.

Author; Year	Type of Study	Characteristics and Size of the Sample	Results	Conclusions
Yuan et al.; 2021 [24]	Prospective, observational	6140 patients with CAD undergoing PCI	Fibrinogen plasma concentrations were positively associated with HbA_1C_ and FBG in CAD patients with and without DM (*p* < 0.001). Elevated fibrinogen plasma concentrations were significantly associated with long-term, all-cause mortality (HR = 1.86; 95% CI: 1.28–2.69; *p* = 0.001) and cardiac mortality (HR = 1.82; 95% CI: 1.15–2.89; *p* = 0.011).	Plasma fibrinogen concentrations in patients with CAD after PCI (especially in patients with DM and pre-DM) were independently associated with the long-term risk of death from all causes and cardiac causes.
Peycheva et al.; 2021 [25]	Observational	153 patients categorised into two groups: with acute ischaemic stroke, and with risk factors but no stroke	Patients with ischaemic stroke had a significantly increased mean plasma fibrinogen concentration (>4 g/L). A significant association between fibrinogen plasma concentrations and the presence of ischaemic lesions on cerebral computed tomography was observed: patients with a fibrinogen concentration > 3.41 g/L showed a 3.29-times increased risk of ischaemic lesions. Analysis of stroke subtypes shows that subjects with undetermined cause of stroke, and subject to atherosclerotic stroke, had significantly higher median fibrinogen plasma concentrations compared to subjects with some other types of strokes. A negative association was established between the clinical evolution of ischaemic stroke subjects and fibrinogen plasma concentrations.	Fibrinogen plasma concentration is a clinically useful biomarker that could characterise acute ischaemic stroke.
Meng et al.; 2021 [26]	Prospective, observational	554 critically ill patients with acute exacerbation of chronic HF	Subjects with plasma fibrinogen concentrations ≥284 mg/mL had a significantly higher risk of death by 185% in the 90-day follow-up period (HR = 2.85; 95% CI: 1.65–4.92, *p* <0.0001).	High-fibrinogen plasma concentrations independently predict mortality in critically ill subjects with acute exacerbation of chronic HF.
Ceasovschih et al.; 2020 [27]	Observational	216 subjects with PAD and 80 subjects without PAD as a control	In subjects with PAD, a significantly higher fibrinogen plasma concentration was demonstrated (417 mg/dL (367–467 mg/dL) vs. 355.5 mg/dL (302.25–362 mg/dL), *p* < 0.0001). Plasma fibrinogen concentration was significantly associated with the risk of PAD (OR = 1.034; 95% CI: 1.005–1.063, *p* = 0.019).	Fibrinogen plasma concentration have a significant value for the presence of PAD.
Samir et al.; 2020 [28]	Observational	64 ICU patients were divided into acute ischemic stroke patients (group I; *n* = 32) and non-stroke patients (group II; *n* = 32)	A significant increase in serum fibrinogen concentrations was noticed in group I (*p* < 0.001) with a cutoff value ≥439 mg/dL showing sensitivity of 92.31%, specificity of 75.36%, and accuracy of 84.34% for stroke occurrence. A cutoff value ≥557 mg/dL showed sensitivity of 85.71%, specificity of 96%, and an accuracy of 93.75% for mortality in this group.	High-serum fibrinogen concentrations in high-risk individuals may be used as a predictor for the occurrence of acute ischemic stroke and mortality from stroke.
Song et al.; 2020 [29]	Prospective, observational	1211 subjects with NSTEMI acute coronary syndromes undergoing PCI	Showed that increased baseline fibrinogen plasma concentrations were an independent predictor of death/nonfatal reinfarction (HR = 1.498; 95% CI: 1.030–2.181, *p* = 0.035).	Fibrinogen plasma concentrations is an independent predictor of death/nonfatal reinfarction in NSTEMI subjects undergoing PCI, and its accuracy is similar to that of the GRACE system.
Liu et al.; 2020 [30]	Retrospective, observational	5237 patients with stable CAD	FBG and HbA_1c_ were positively associated with fibrinogen plasma concentrations in overall CAD subjects, either with or without DM (all *p* < 0.001). High fibrinogen plasma concentrations were independently associated with MACEs after adjusting for confounding factors (HR = 1.57; 95% CI: 1.26–1.97, *p* < 0.001).	Fibrinogen plasma concentrations were associated with FBG and HbA_1c_ in stable CAD subjects. Moreover, increased fibrinogen plasma concentrations were independently associated with a risk of MACEs in CAD subjects, especially among those with DM and pre-DM.
Jiang et al.; 2019 [31]	Prospective, observational	6293 patients undergoing PCI	The 2-year all-cause mortality rate was 1.2%. Patients with higher plasma fibrinogen concentrations died more frequently than those with low or moderate levels (1.7% vs. 0.9% and 1.7% vs. 1.0%, respectively; *p* = 0.022). Fibrinogen was significantly associated with risk of all-cause mortality (HR = 1.339; 95% CI: 1.109–1.763, *p* = 0.005).	High-fibrinogen plasma concentrations were associated with a worse prognosis in subjects after PCI.
Zhang et al.; 2019 [32]	Prospective, observational	411 ACS patients undergoing PCI (103 subjects with DM and 308 subjects with non-DM)	Patients with DM had higher plasma concentrations of fibrinogen than patients without DM (3.56 ± 0.99 mg/dL vs. 3.34 ± 0.80 mg/dL, *p* < 0.05). HbA_1c_ and FBG were significantly positively correlated with fibrinogen in patients with DM, but not in subjects without DM (all *p* < 0.05). Increased plasma fibrinogen concentration was significantly associated with a higher risk of MACE only in patients with DM (HR = 7.783; 95% CI: 1.012–59.854, *p* = 0.049).	Fibrinogen was positively associated with glucose metabolism in DM populations with ACS. Moreover, elevated baseline fibrinogen plasma concentrations may be an important and independent predictor of MACEs following PCI, especially amongst those with DM.
Chen et al.; 2018 [33]	Cross-sectional	1096 T2DM patients	Patients with PAD had higher serum fibrinogen concentrations than non-PAD group (*p* < 0.001). Higher fibrinogen quartiles were positively related with the development of PAD— Tercile 2 (3.02–3.65 g/L): OR = 1.993; 95% CI: 1.322–3.005, *p* < 0.001; Tercile 3 (3.66–4.55 g/L): OR = 2.469; 95% CI: 1.591–3.831, *p* < 0.001; Tercile 4 (≥4.56 g/L): OR = 2.942; 95% CI: 1.838–4.711, *p* < 0.001.	Serum fibrinogen concentrations were an independent risk factor for PAD in patients with T2DM.
Gao et al.; 2017 [34]	Observational	418 males with myocardial infraction who were under 35 years old	Positive correlation between plasma fibrinogen concentration and GS was found (*p* < 0.001). The best cut-off level for plasma fibrinogen concentration predicting the severity of coronary stenosis was 3.475 g/L (sensitivity 64%; specificity 70%). Plasma fibrinogen concentration was also independently associated with high GS (OR = 2.173; 95% CI: 1.011–4.670, *p* = 0.047).	Plasma fibrinogen concentration is significantly associated with the presence and severity of coronary artery stenosis in men under 35 years of age with MI.
Tabakci et al.; 2017 [35]	Observational	134 subjects with stable CAD	Strong correlation between fibrinogen plasma concentrations and the SS (r = 0.535, *p* < 0.001). Fibrinogen plasma concentrations higher than 411 mg/dL had a sensitivity of 75% and a specificity of 64% in the prediction of high SS. Plasma fibrinogen concentrations were an independent predictor for high SS in subjects with stable CAD (OR = 1.01; 95% CI: 1.01–1.02, *p* < 0.001).	Plasma fibrinogen concentrations were independently associated with severity and complexity of CAD.
Yang et al.; 2017 [36]	Prospective, observational	1466 subjects with T2DM and angiographically proven stable CAD	Patients who had high plasma fibrinogen concentration (≥3.51 g/L) had a significantly higher risk of CVD by 102% (HR = 2.02; 95% CI: 1.11–3.68, *p* = 0.049).	Elevated fibrinogen plasma concentrations were independently associated with higher risk of CVD.
Peng et al.; 2017 [37]	Retrospective, observational	2253 patients with acute coronary syndrome confirmed by coronary angiography	Cumulative survival curves indicated that the risk of all-cause death increased with increasing plasma fibrinogen concentration (mortality rates for Tercile 1 vs. Tercile 2 vs. Tercile 3 = 6.6% vs. 10.8% vs. 12.3%, *p* < 0.001). Similar trends were observed for CVD death, although the differences between terciles were not statistically significant (cardiac mortality rates for Tercile 1 vs. Tercile 2 vs. Tercile 3 = 4.6% vs. 6.3% vs. 6.4%, *p* = 0.206). HR for all-cause mortality and cardiac mortality across terciles (3 vs. 1) of fibrinogen: 1.96; 95% CI: 1.39–2.77 and 1.47; 95% CI: 1.03–2.10.	Plasma fibrinogen concentrations at admission were independently associated with risk of death among subjects with acute MI.
Kunutsor et al.; 2016 [38]	Prospective with meta-analysis	1773 men free of HF or cardiac arrhythmias who recorded 131 SCD for 22 years of follow-up	Men who experienced SCD had a higher plasma fibrinogen concentration (2.93 g/L (92.61–3.30) vs. 3.19 g/L (2.87–3.57), *p* < 0.0001). Fibrinogen was log-linearly associated with risk of SCD. Hazard ratio for SCD per 1 standard deviation higher baseline loge fibrinogen was 1.32 (95% CI: 1.11–1.57). Meta-analysis of three cohort studies was showed that fully adjusted the relative risks for SCD per 1 standard deviation higher baseline and long-term fibrinogen plasma concentrations were 1.42 (95% CI: 1.25–1.61) and 2.07 (95% CI: 1.59–2.69), respectively.	Fibrinogen plasma concentrations were positively, log-linearly, and independently associated with the risk of SCD.
Kotbi et al.; 2016 [39]	Prospective, observational	120 subjects: 30 non-DM and with CAD, 30 with DM and CAD, 30 non-CAD with DM, and 30 healthy subjects	The plasma fibrinogen concentration increased in parallel with the CVD risk (*p* = 0.0001); there was also a significant correlation between the plasma fibrinogen concentration and the clinical and para-clinical CAD severity (respectively *p* = 0.005 and *p* = 0.0001).	Plasma fibrinogen concentrations were positively and significantly associated with the CAD severity.
Peng et al., 2016 [40]	Observational	3020 subjects with CAD confirmed by coronary angiography	Cumulative survival curves showed that the risk of all-cause mortality was significantly higher in subjects with plasma fibrinogen concentrations ≥3.17 g/L vs. those with < 3.17 g/L (mortality rate, 11.5% vs. 5.7%, *p* < 0.001); and cardiac mortality rate—5.9% vs. 3.6%, *p* = 0.002). Plasma fibrinogen concentrations remained independently associated with all-cause mortality after adjustment for multiple CVD risk factors (HR = 2.01; 95% CI 1.51–2.68, *p* < 0.001).	Plasma fibrinogen concentrations were independently associated with the mortality risk in CAD patients.
Peng et al.; 2016 [41]	Observational	3020 patients with CAD confirmed by coronary angiography	Mortality rates for subjects with CAD and those in the stable CAD and unstable CAD groups exhibited an overall rising trend as fibrinogen plasma concentrations increased (all *p* < 0.05). Fibrinogen plasma concentrations were independently associated with the risk of death in CAD subjects, as well as those in the stable CAD and unstable CAD groups (CAD, HR = 1.40; 95% CI: 1.16–1.68; stable CAD, HR = 1.86; 95% CI: 1.24–2.79 and unstable CAD, HR = 1.42; 95% CI: 1.06–1.90). In the acute MI group, however, no independent correlation was observed between fibrinogen plasma concentrations and mortality.	The different proportions of subtypes of CAD affected the correlation between fibrinogen plasma concentrations and the clinical prognosis of subjects with CAD.
Zhang et al., 2014 [42]	Observational	2288 new-onset subjects undergoing coronary angiography with angina pain	Subjects with high GS had significantly increased fibrinogen plasma concentrations (*p* < 0.001). Plasma fibrinogen concentrations were independently associated with high GS (OR = 1.275; 95% CI: 1.082–1.502, *p* = 0.004) after adjusting for potential confounders. The risk of stenosis (≥75%) was increased with the elevated plasma fibrinogen concentrations: Tercile 2 (2.83–3.38 g/L)—OR = 1.112; 95% CI: 0.887–1.395, *p* = 0.365. Tercile 3 (> 3.38 g/L)—OR = 1.939; 95% CI: 1.484–2.533, *p* < 0.001.	Higher fibrinogen plasma concentrations were independently associated with new-onset atherosclerosis in the coronary arteries
Hong et al.; 2014 [43]	Observational	373 subjects with DM and angina pectoris	Plasma fibrinogen concentration was an independent predictor of a high GS for DM subjects (OR = 1.40; 95% CI: 1.04–1.88, *p* = 0.026) after adjusting for traditional risk factors of CAD.	Plasma fibrinogen concentrations appeared to be an independent predictor for the severity of CAD in DM subjects
Bosevski et al.; 2013 [44]	Prospective, observational	62 patients with T2DM and PAD Follow-up: 36 months	Linear regression analysis defined plasma fibrinogen concentrations as a predictor for endpoint value of ankle-brachial index (β = 0.469, *p* = 0.007).	Plasma fibrinogen concentrations can be used to evaluate the progression of PAD in subjects with T2DM.

**Table 3 ijms-23-00193-t003:** Fibrinogen molecular modifications and cardiovascular risk. CVD - cardiovascular disease; CAD—Coronary artery disease; MACE—major adverse cardiovascular events.

Gene	Polymorphism/Mutation	Effect on Cardiovascular Risk	Bibliography
*FGG*	γ′ fibrinogen	↑ myocardial infraction ↑/↔ CVD ↓ venous thromboembolism and ischemic stroke	[67,74,76,77]
	rs7681423 and rs1049636	↔ CVD	[67]
	rs2066865	↑ microvascular thrombosis	[75]
*FGB*	-455 G/A	↔ venous thromboembolism ↓ venous thromboembolism (Caucasians) ↑ ischemic stroke (Asian) ↔ ischemic stroke (Caucasians and children) ↑ cerebral infarction ↑ CAD ↑ cardioembolic stroke	[84,85,86,87,88,89]
	-148 C/T	↔ venous thromboembolism ↑ ischemic stroke (Asians and Caucasians) ↑ cerebral infarction ↑ CAD ↑ MACE	[84,85,86]
	-1420 (AG + AA) and -148 (CT + TT)	↑ lower extremity deep venous thrombosis	[95]

## Data Availability

Not applicable.

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
