# Peer review of "Fibrinogen and Atherosclerotic Cardiovascular Diseases—Review of the Literature and Clinical Studies"

_ijms, 2021, doi:10.3390/ijms23010193_

Round 1

Reviewer 1 Report

The authors of the manuscript focused on the role of fibrinogen in cardiovascular diseases. This is a current topic in internal medicine. The association between fibrinogen and coronary artery disease has long been a concern. Fibrinogen and fibrin have a major role in multiple biological processes in addition to hemostasis and thrombosis, i.e., fibrinolysis (during which the fibrin clot is broken down), matrix physiology (by interacting with factor XIII, plasminogen, vitronectin, and fibronectin), wound healing, inflammation, infection, cell interaction, angiogenesis, tumour growth, and metastasis. The manuscript is well structured and detailed review article, some facts need to be supplemented and corrected.

Page 6, lines 66-67: Normal level ranges may vary slightly among different laboratories. Some laboratories use different measurements or different reagents. This is important for authors to state.

Pages 2-4, lines 63-103: Fibrinogen is a 340 kDa glycoprotein comprising pairs of three polypeptide chains termed Aα, Bβ, and γ. Fibrinogen has a trinodular structure with a central nodule (E-domain) that contains the N-terminus of each chain and two lateral globular domains (D-domains) that contain the C-terminus of Bβ- and γ-chains. The E-domain is linked to the two D-domains by a coiled-coil triple helix structure. This is very important to describe the fibrinogen domains and molecular weight in detail. Authors should cite the manuscript in which it was described: Semin Thromb Hemost. 2016 Jun;42(4):455-8. doi: 10.1055/s-0036-1581104.

Page 16, lines 478-485: The authors report polymorphisms in the FGB chain that lead to thrombotic complications. The authors should state that in addition to higher fibrinogen levels, low fibrinogen levels and mutations in the FGB chain were associated with a thrombotic phenotype. In the absence of fibrinogen or at low levels, the small amount of thrombin usually formed remains longer in the circulation as no or less sequestering on circulating fibrinogen occurs (ie antithrombin function of fibrin is impaired). Besides, thrombin generation has been shown to be increased in the plasma of patients with low levels of fibrinogen. Authors should cite the manuscript in which it was described: Thromb Res. 2020 Apr;188:1-4. doi: 10.1016/j.thromres.2020.01.024

Pages 12-13, lines 327-368: γC region (residues 143–411) forms a single globular domain and has been conducted to play a critical role in fibrinogen assembly and secretion, both in vivo and in cultured cells. Several mutations have been described in this area that leads to hypofibrinogenemia associated with a thrombotic phenotype. It is therefore this claim to discuss and cite the manuscript in which it was described:  Congenital hypofibrinogenemia associated with a novel heterozygous nonsense mutation in the globular C-terminal domain of the γ-chain (p.Glu275Stop). J Thromb Thrombolysis. 2020 Jul;50(1):233-236. doi: 10.1007/s11239-019-01991-x.

Figures and tables in the text are very clearly written.

I have to say that with these 100 references. Several references are from the last 5 years.

Author Response

Thank you for your comments and additions. We fully agree with them.

1.1 We have introduced information on the possibility of differences in the correct norm of fibrinogen concentration in different laboratories

1.2 We added information on molecular weight and the structure of fibrinogen.

1.3 We have added the appropriate paragraph and the suggested literature to the text.

1.4 We have added the appropriate paragraph and the suggested literature to the text.

1.5 Thank you very much

1.6 We have added the proposed literature and also added a short paragraph on the important role of fibrinogen in COVID-19.

Reviewer 2 Report

The paper is interesting and well written. I suggest to  add as references papers by Murdaca et al concerning endothelial dysfunction and free radicals in accelerated atherosclerosis

Author Response

Thank you for your review.

2.1. We have added the proposed literature.

Round 2

Reviewer 1 Report

The presented manuscript has been corrected in response to the suggestions. The authors have followed the recommendations of the reviewer. After the revision, the provided data and addition of the results became more clear. I would like to thank the authors for resubmitting the manuscript and explaining the obscure points from the previous version.